# Complement Properdin Regulates the Metabolo-Inflammatory Response to a High Fat Diet

**DOI:** 10.3390/medicina56090484

**Published:** 2020-09-22

**Authors:** Rόisín C. Thomas, Ramiar Kheder, Hasanain Alaridhee, Naomi Martin, Cordula M. Stover

**Affiliations:** 1Department of Respiratory Sciences, University of Leicester, Leicester LE1 9HN, UK; rct21@le.ac.uk (R.C.T.); ramiarkheder@gmail.com (R.K.); hasanain.alarithy@gmail.com (H.A.); naomi.martin@dmu.ac.uk (N.M.); 2Faculty of Health and Life Sciences, De Montfort University, Leicester LE1 9BH, UK

**Keywords:** diet, complement, mouse model, C5L2, CD36

## Abstract

*Background and objectives:* Overnutrition leads to a metabolic and inflammatory response that includes the activation of Complement. Properdin is the only amplifier of complement activation and increases the provision of complement activation products. Its absence has previously been shown to lead to increased obesity in mice on a high fat diet. The aim of this study was to determine ways in which properdin contributes to a less pronounced obese phenotype. *Materials and Methods:* Wild type (WT) and properdin deficient mice (KO) were fed a high-fat diet (HFD) for up to 12 weeks. *Results:* There was a significant increase in liver triglyceride content in the KO HFD group compared to WT on HFD. WT developed steatosis. KO had an additional inflammatory component (steatohepatitis). Analysis of AKT signalling by phosphorylation array supported a decrease in insulin sensitivity which was greater for KO than WT in liver and kidney. There was a significant decrease of C5L2 in the fat membranes of the KO HFD group compared to the WT HFD group. Circulating microparticles in KO HFD group showed lower presence of C5L2. Expression of the fatty acid transporter CD36 in adipose tissue was increased in KO on HFD and was also significantly increased in plasma of KO HFD mice compared to WT on HFD. CD36 was elevated on microparticles from KO on HFD. Ultrastructural changes consistent with obesity-associated glomerulopathy were observed for both HFD fed genotypes, but tubular strain was greater in KO. *Conclusion:* Our work demonstrates that complement properdin is a dominant factor in limiting the severity of obesity-associated conditions that impact on liver and kidney. The two receptors, C5L2 and CD36, are downstream of the activity exerted by properdin.

## 1. Introduction

Complement is well known for its role in defence against pathogens. More recently it has also emerged as a component in the pathology of chronic inflammatory conditions such as cardiovascular disease, neurodegenerative disease, and autoimmune disease as well as in pain and ischemia. The role of complement in metabolism is one such new frontier [1].

Obesity is associated with low grade inflammation [2]. The alternative pathway of complement is activated by chylomicrons (transporters of dietary lipids). This leads to the production of C3a desArg (identical with acetylation stimulation peptide, ASP) which is a proposed ligand for the C5L2 receptor. The C5L2 receptor is involved in the synthesis of triglycerides from fatty acids that are transported into adipocytes [3]. This activity has been described as a functional feature of adipose tissue. It is distinct from the well-known role of the alternative pathway of complement which is to amplify the complement system in response to molecular patterns that are associated with pathogens or danger [4]. Patients with hypertriglyceridemia have chylomicrons in their serum even when fasting [5].

Factor D is a serine protease (originally known as adipsin) that is unique to the alternative pathway and has been found to be significantly lower in the sera of obese mice (compared to background matched controls) [6]. This implies a direct relationship of alternative pathway activity to the metabolic use of fatty acids [7]. ASP stimulates lipogenesis in adipocytes in synergy with insulin: By inhibiting the release of fatty acids from lipolysis and re-esterifying free fatty acids, ASP led to increased lipid storage in adipocytes [8]. C5L2 expression in liver was shown to be protective against steatosis [9]. C5L2 KO mice had greater lipid content in their livers compared to wildtype mice when placed on a high fat high sucrose diet (58% kcal fat) for twelve weeks [10].

In a previous study, using properdin deficient mice generated by W. Song, we found that male properdin deficient mice on a diet of 45% kcal fat were more obese than their wildtype male littermates. While the body fat mass was significantly increased in high fat diet fed properdin deficient mice, there was no hepatomegaly. There was hyperinsulinemia in both groups. Properdin deficient mice showed elevated triglycerides, and significantly reduced non esterified fatty acids (NEFA) compared to wildtype mice [11]. The mechanism underlying the metabolic phenotype in properdin deficient mice was not entirely elucidated in this work. In vivo and in vitro studies have related ASP levels to measures of metabolic output but did not analyse concomitantly the expression of the ASP receptor, C5L2 [12]. Similarly, C5L2 KO mice were not analysed for their complement activity profile [13].

The C5L2 (C5aR2 or GP77) receptor is a seven transmembrane receptor that is expressed in a diverse array of cell types and tissues including immune cells, adipose tissue and the liver [10]. It is similar in structure to the C5a complement receptor (C5aR1) but unlike the C5aR1 it does not couple to (and therefore does not signal via) Gα-proteins. It was initially thought to function only as a decoy receptor, due to its ability to bind C5a and C5a desArg (a hydrolytic product of C5a) and subsequently undergo internalisation, leading to degradation of those ligands [14]. The function of C5L2 was later questioned when evidence was found to show that it couples to beta arrestins upon stimulation with C5a. It was also found that it can dimerize with C5aR1. This led sequentially to internalisation of C5aR1, the recruitment of the endocytosis-related protein adaptin-2 and subsequent downstream ERK signalling [15]. There have been conflicting reports on whether this interaction with C5aR1 is pro or anti-inflammmatory [16].

Beta arrestin coupling and internalisation of the C5L2 receptor have also been shown in response to ASP [17,18,19]. (There has been much debate surrounding the issue of whether C3a desArg asserts its effects by binding directly to C5L2. For a comprehensive review see Zhang et. al, 2017 [16] which describes animal and clinical data that supports a role for both receptor and ligand in metabolism and obesity.)
In vitro ASP stimulation of C5L2 led to activation of Phospholipase C, protein kinase C and phosphorylation of AKT. Additionally, its stimulation activated downstream signalling via MAPK/ERK1/2, and cytosolic phospholipase A2 (cPLA2) resulting in triglyceride synthesis [20]. ASP has been shown to stimulate glucose transport [21] and inhibit hormone sensitive lipase [12].

A deletion of either the C5L2 receptor gene, or the gene for C3 (the precursor of C3a desArg, ASP) resulted in a delay in postprandial triglyceride clearance and uptake of free fatty acids in mice [8,22]. However, administration of exogenous C3a desArg to C3KO mice restored the time of clearance and uptake to normal [23,24]. C3a, modified by the action of plasma carboxypeptidase N to become desarginated C3a desArg, acquires affinity to C5L2, a receptor found on adipocytes, where it mediates uptake of dietary fatty acids.

Inflammation and other stimuli promote the generation of ASP via the alternative complement pathway. These include stimulation in vitro of adipocytes by the presence in plasma of chylomicrons, dietary lipoproteins and insulin [21,25,26]. Long term consumption of a Western style diet leads to hypertrophy and hyperplasia of adipocytes, significant producers of components of the alternative pathway [4], and lipolysis of stored fat when insulin sensitivity is compromised. At this stage C5L2/C3a desArg signalling could become impaired [27].

Increase of adipose tissue and development of fatty liver disease due to consumption of a Western Diet correlates with increased complement activation in fat and liver, respectively. The increased metabolic demand on adipose tissue alters its role: It becomes an organ which contributes locally and systemically to the diet associated inflammatory reaction via macrophage activation and adipokine release [28]. The extent of inflammation, which includes complement activation, impacts on the development of hepatic insulin resistance [29]. The levels of ASP correlated significantly with HOMA-IR (Homeostatic Model Assessment of Insulin Resistance) in a group of patients with non-alcoholic fatty liver disease (NAFLD)/non-alcoholic steatohepatitis (NASH) [30], when oxidative stress is an increasingly important determining factor for the local tissue environment [31]. Increased release of microparticles contributes to the progression of NAFLD to NASH [32]. Microparticles are biologically active nanovesicles that are shed from activated cells and can act as biomolecular vehicles [33]. They are involved in inflammatory processes and disease pathophysiology. Liver, skeletal muscle and adipose tissue are classically regarded as insulin sensitive organs, but insulin also signals in the kidney via PI3K/pAKT pathway [34]. Indeed, an inflammatory and metabolic crosstalk is described in obesity between adipose tissue, kidney, and liver [35].

CD36 (known also as Fatty acid translocase, platelet glycoprotein IV or glycoprotein IIIb) is a membrane receptor that is expressed on many cell types. These include adipocytes [36], liver cells [37,38], kidney cells [39], microvascular endothelial cells [40], macrophages [41] and platelets [42]. Ligands that bind to the receptor include long chain fatty acids, oxidized low density lipoproteins and phospholipids [43]. Described as both a scavenger receptor and a fatty acid translocase, two well-described functions of CD36 are removal of oxidised lipids and uptake of fatty acids from the blood [44]. CD36 has been associated with the pathogenesis of type 2 diabetes and metabolic syndrome [45,46]. A connection between CD36 and properdin has been suggested but not shown [11].

This study analysed the complex metabolic response of properdin deficient and wildtype littermates to obesity. We measured the extent of chronic inflammation, hepatic and renal involvement, as well as regulation of C5L2 and CD36.

## 2. Methods and Materials

### 2.1. Mice and Experimental Design

Approval of the programme of work (Complement properdin in immunity and inflammation) was granted by the institutional Animal Welfare and Ethics Subcommittee (item AWERB/15/24) and by the Secretary of State of the UK Home Office (license P43308E3B) on 11 July 2016. The severity of the protocol used in this paper was classed as mild.

Properdin deficient mice and their congenic wildtype controls were taken from a colony maintained by intercrossing heterozygous x wildtype mice. Only male mice were used for this work, so littermates (because of x-chromosomal linkage of the *Cfp* gene) could be used [47]. Mice were housed in groups in ventilated cages at 21 °C, 50% humidity, with 12/12 h light/dark cycle, and had ad libitum access to food and water. When mice were 3–4 months of age, a Western diet (5TJN, calories provided: protein 15.8%, fat 39.1%, carbohydrates 45.1%) (High fat diet, HFD group) or maintenance diet (5LF2, calories provided: protein 16.8%, fat 6.6%, carbohydrates 76.6%) from TestDiet (International Product Supplies, London, UK) (normal diet, ND group) were fed. Gnawing blocks were added to the cage floor covered with corn cob as bedding material; nesting material (sizzle pet) was made from recycled paper. There was equal environmental enhancement for all. Consumption of high fat diet decreased towards end of study but there was no difference in food consumption between genotypes. At the end of study, blood was withdrawn under terminal anaesthesia. In total, 90 mice were used.

### 2.2. Serum Measurements

Activities of liver transaminases aspartate aminotransferase (AST) and alanine aminotransferase (ALT) were determined in serum samples following the manufacturer’s instructions (Abcam, Cambridge, UK). NEFA was measured using NEFA Assay kit from WAKO Chemicals GmbH (Neuss, Germany). Triglyceride colorimetric assay kit was from Cayman Chemical (Ann Arbor, Michigan, USA). Malondialdehyde was measured by immunoassay kit (Abcam and Biorbyt, Cambridge, UK). Adiponectin was measured using ELISA Mouse Adiponectin/Acp30 (R&D Systems, Abingdon, UK) using serum in 2000-fold dilution as suggested by the manufacturer. HbA1c was measured in 1:10 diluted serum samples using mouse glycated Hemoglobin A1c ELISA kit (CUSABIO, Wuhan, China). Endotoxin was measured by Limulus amebocyte lysate (LAL) method in serum diluted 1: 50 (Pierce^TM^ LAL chromogenic endotoxin quantification kit, Thermo Scientific, Waltham, MA, USA). Murine Interleukin (IL)-6 ELISA kit (Peprotech, London, UK) was used according to manufacturer’s instructions. Serum dilution was 1:10. C5a was measured by sandwich ELISA using purified rat anti-mouse C5a (BD Pharmingen), serum samples were diluted 1:20 and purified recombinant mouse C5a (BD Pharmingen) was used to generate a standard curve. Complement activity assays for alternative and classical pathways were performed following a published method [48]. In order to compare activities between separate assays, a commercially available mouse serum (Life Technologies, Carlsbad, CA, USA, Cat No. 10410) was tested in parallel and activities expressed as percentage to this serum. Heat inactivation of sera to prove heat lability of the complement assays was performed at 56 °C for 30 min. 10 mM ethylenediaminetetraacetic acid (EDTA) was added to prove ion dependence of the activations.

### 2.3. Measurement of Liver Triglyceride Content

50 mg of liver tissue was finely chopped and placed in a 1.5 mL reaction tube prior to addition of 20 volumes of 2:1 chloroform:methanol (*v*/*v*). Samples were then vortexed intermittently at room temperature for 5 min. Supernatants were then placed in a fresh reaction tube. 0.2 volumes (of the total liquid volume) of water was then added and the samples were vortexed again. The solution was then sedimented at 1000× *g* for 10 min. The upper layer was then discarded, and the lower layer was placed into a fresh glass tube. The solvent was then evaporated from the glass tube using a vacuum concentrator. The lipids were then resuspended in 100 µL ethanol. Triacylglycerides were then quantified using a LabAssay^TM^ Triglyceride kit (Fujifilm Wako Pure Chemical Corporation, Osaka, Japan) according to the manufacturer’s instructions.

### 2.4. Histology and Immunohistochemistry

Tissues were fixed in 10% neutral buffered formalin and embedded in paraffin. 5-micron sections were stained with hematoxylin eosin. Anti CD36 was from Novus Biologicals (NB400-145).

### 2.5. Phosphorylation Arrays

To measure phosphorylation of proteins downstream of the insulin receptor we used the RayBio^®^ C-Series Human and Mouse AKT Pathway Phosphorylation Array C1 kit. 50 mg of tissue from nonfasted experimental animals (*n* = 6 per group) was lysed in 1 mL lysis buffer containing phosphatase and protease inhibitors. Samples were pooled by group and total protein concentration (determined using Pierce 660 nm Protein Assay Reagent kit, Thermo Fisher Scientific) was adjusted to 1000 µg/mL. Chemiluminescent signals on the membranes (all membranes at the same time) were then imaged using a ChemiDoc touch imaging system (Bio-Rad Laboratories Ltd., Watford, UK). Raw densitometry data was extracted using ImageLab 6.0 software (Bio-Rad) using an extraction circle of the same dimensions for each array spot. Negative control spots were used to subtract background responses. To normalise the array data the membrane of the wild type, normal diet group was chosen as the reference array and the signal fold expressions of the other array membranes compared to this were calculated using the algorithm in the manufacturer’s manual. Signal fold changes were then converted to percentage change compared to wild type on normal diet e.g., a 1.40-fold increase in the signal intensity is a 40% increase.

### 2.6. Enzyme Linked Immunosorbent Assays (ELISA) for C5L2 and CD36

To determine the concentration of CD36 in plasma using RayBio^®^ Mouse CD36 ELISA kit (RayBiotech Life, Peachtree Corners, GA, ELM-CD36), plasma samples were diluted 1:10. Plasma membranes of adipose tissues were isolated as previously described [49], protein content was adjusted to 100 μg/100 μL (100 μg/well) and concentrations of C5L2 were determined using EIAab C5L2 receptor ELISA kit (EIAab, Wuhan, China, E15353m).

### 2.7. Antibodies

The following antibodies were used for microparticles: CD36 antibody (Novus Biologicals, Littleton, CO, USA) and Texas Red conjugated immunoglobulins, Annexin V-fluorescein isothiocyanate (AnV-FITC) for phosphatidylserine (BD Biosciences, San Jose, CA, USA 556420), tissue factor CD142-phycoerythrin (PE) (BD Biosciences 550312), C5L2 Alexa Fluor^®^ 647-Conjugated mouse monoclonal antibody (R&D Systems, Abingdon, UK, IC4729R), mouse IgG2b Isotype Control (MPC-11) Alexa Fluor^®^ 647-Conjugated mouse monoclonal antibody (Novus Biologicals. NBP2-2722B). 2 microglobulin mouse monoclonal antibody (Santa Cruz Biotechnology sc-46697), Smooth muscle alpha actin antibody (Santa Cruz Biotechnology, Dallas, TX, USA), CD36 Rabbit polyclonal antibody (Novus bio NB400-145) and Swine Anti-Rabbit horseradish peroxidase (Dako, Glostrup, Denmark, P0399) were used for Western blotting.

### 2.8. Western Blotting

Equal amounts of tissue were placed into ice cold 1% (*v*/*v*) NP40 lysis buffer with protease (SIGMAFAST^TM^ Protease inhibitor cocktail tablets. Sigma Aldrich, St Louis, MI, USA, S8830) and phosphatase inhibitors (Broad Spectrum Phosphatase inhibitor cocktail. Boster Biological Technology, Pleasanton, CA, USA, AR1183) and homogenised on ice then incubated on ice for 30 min prior to sedimentation (10,000× *g* for 10 min at 4 °C). The supernatant was then removed and sedimented for a further 10 min as above. Protein concentrations were determined using a Pierce™ 660 nm Protein Assay Kit (Thermo Fisher Scientific) and samples were diluted to equal concentrations. Samples were then diluted 1:1 with 2× Laemmli sample buffer. Urine samples were simply diluted 1:1 with sample buffer. For isolation of the plasma membrane of adipocytes the protocol described in [49] was followed.

Proteins were boiled for 5 min and 20 µL of each sample were separated by Sodium dodecyl sulfate polyacrylamide gel electrophoresis (SDS PAGE) before being transferred to a nitrocellulose membrane via tank transfer (250 mA for 1 h in transfer buffer: 25 mM Tris, 192 mM Glycine, 10% (*v*/*v*) Methanol). Membranes were then stained with Ponceau S stain to check that protein loading was equal.

Membranes were then blocked for 1 h in phosphate buffered saline containing 5% (*w*/*v*) skimmed milk and 0.1% (*v*/*v*) Tween 20 before incubation overnight at 4 °C in PBS/T-5% (*w*/*v*) milk containing primary antibodies.

Membranes were then washed in 3 changes of PBS/T for 10 min before incubation with PBS/T-5% (*w*/*v*) milk containing secondary antibodies conjugated to horseradish peroxidase for 1 h. This was followed by a further wash in 3 changes of PBS/T for 10 min before incubation in Novex^®^ ECL Chemiluminescent Substrate (Thermo Fisher Scientific) for 1 min. Membranes were then either exposed to X ray film or imaged on a ChemiDoc touch imaging system (Bio-Rad).

### 2.9. Flow Cytometry/Microparticles

Plasma was spun at 2500× *g* for 15 min 18–20 °C. The top platelet free phase was used for analysis.

Aliquots of microparticles were incubated alone, with AnV-FITC, CD142-PE, C5L2-APC647, CD36 alone or CD36 plus Texas Red conjugated secondary antibody in the dark for 25 min at room temperature. Microparticles were then resuspended in 0.2μm-filtered phosphate buffered saline and acquired using flow cytometry. All flow cytometric acquisition and analysis was performed using a BD Accuri™ C6 Plus and software (both BD Biosciences, Oxford, UK). Microparticles were acquired using a gating defined by size-calibrated fluorescent Megamix™ beads according to the manufacturer’s instructions (BioCytex, Marseille, France) and according to a standardized calibrated-bead strategy [50]. All samples were acquired using the slowest flow rate for a fixed time period of 120 s.

### 2.10. Electron Microscopy

Small pieces of kidney cortex (1 mm^3^) were fixed overnight in 2.5% (*v*/*v*) glutaraldehyde/2% (*v*/*v*) formaldehyde in 0.1 M sodium cacodylate buffer (pH7.3), and then taken through several sodium cacodylate buffer washes. Tissue was further fixed in 1% (*w*/*v*) osmium tetroxide with 1.5% (*w*/*v*) potassium ferricyanide in buffer for 90 min, followed by several washes in de-ionised water. After a brief wash in 30% (*v*/*v*) ethanol, the tissue was tertiary fixed in 1% (*w*/*v*) uranyl acetate in 30% ethanol for 1 h in the dark. Dehydration was continued through a series of 50%, 70%, 90% and several 100% ethanol washes. After two propylene oxide exchanges, the tissue was gradually infiltrated through a propylene oxide/TAAB 812 epoxy resin gradient. The samples were left to infiltrate in 100% resin overnight. Tissue went through 3 more changes of 100% resin over the course of the next day before polymerisation at 60 °C for 16 h.

Ultramicrotomy was performed on a Leica EM UC7. Semi thick (400 nm) sections were collected onto glass slides, dried, and stained with 1% (*w*/*v*) toluidine blue/1% (*w*/*v*) borax solution. Thin sections (70 nm) were collected onto copper mesh grids and stained with 2% aqueous uranyl acetate for 30 min followed by 5 min in Reynold’s lead citrate. TEM grids were viewed on a JEOL JEM-1400 TEM with an accelerating voltage of 100 kV. Digital images were collected using a Megaview III digital camera with iTEM software (or an EMSIS Xarosa digital camera with Radius software.)

### 2.11. Statistical Analysis

Gaussian distribution of measurements was assumed. Data were presented as means ± SD and analysed by unpaired *t*-test or one-way ANOVA where appropriate using Prism Pad 6.

## 3. Results

### 3.1. Changes of Biochemical Markers

Properdin deficient (KO) and wildtype (WT) mice received Western diet for 10 or 12 weeks at 3–4 months of age. Some mice received the diet for 8 weeks. The high fat diets (40% kcal from fat compared to 6% in maintenance diet) were palatable and there was no difference in intake between the genotypes. The extent of gain in body weight and in epididymal fat pad weight among high fat diet fed WT and KO mice was variable. Mice gained between 4 and 13.6 g total weight and fat pad as % of body weight was between 3.1 and 5.7. Compared to congenic WT mice on high fat diet, KO mice, however, appeared increasingly metabolically strained: they had lower adiponectin levels -typical marker of insulin resistance [51], and elevated HbA1c (while non fasting blood glucose was normal, data not shown), showed significantly higher levels of IL-6, an adipocytokine typically elevated in obesity [52] and lipid peroxidation products (malondialdehyde) as well as significant elevations of NEFA, lipids and endotoxins (Table 1).

ALT, not AST , was found elevated, consistent with the development of steatosis, which we assessed histologically (Figure 1).

In previous work we had surmised that properdin directly (independent of its role in serum) inhibited free fatty acid uptake by adipocytes but one cannot exclude that fatty acids were simply bound by the basic nature of recombinant properdin used in the in vitro assay [11]. We assayed serum complement activities in mice receiving the different diets: A functional complement activity test that detects deposited C9 in wells coated with alternative pathway activator *Salmonella enteritidis* lipopolysaccharide showed, as expected, a significant impairment of alternative pathway activity in serum from KO mice, as we have shown before using a rabbit erythrocyte lysis assay [47]. The normal activity in serum from WT mice was significantly decreased when serum from mice fed a high fat diet was tested. This is consistent with increased activation in vivo, leading to consumption of the complement components necessary to yield deposited C9 (in the form of C5b-poly C9) on the plate. The classical pathway, by contrast did not show this level of consumption (Table 2).

Because KO mice are impaired in their ability to sustain alternative pathway activity in the absence of stabilisation of C3 and C5 convertases, the generation of the split product C3a desArg, ligand for C5L2, is likely to be affected, when demand is increased. C5a, a surrogate marker for this compromised activity, was significantly lower in high fat diet fed KO mice than in WT (Table 1). A metabolic effect of alternative pathway activation for uptake of NEFA has been described [53].

### 3.2. Insulin Signalling in Liver Tissue

Livers of experimental mice showed histological changes consistent with steatosis in properdin WT mice and steatohepatitis in KO mice (Figure 2a). There was a significant increase in triglyceride content in the WT HFD group compared to controls on ND. The triglyceride content was higher still in the KO group with a significant difference between the genotypes (Figure 2b).

The metabolic response to overnutrition engages the AKT signalling pathway. The WT HFD group showed an increase in AKT phosphorylation compared to control of 162.5% in the liver (Table 3). The increase in AKT phosphorylation in the liver of KO was only 71% with a difference between the genotypes of 91%. The WT HFD group showed an increase in ERK1/2 phosphorylation of 176% in the liver compared to control whereas the KO HFD group showed an increase of 238% with a difference of 62% difference between the genotypes. There was also an increase in phosphorylation of PRAS40 in both HFD groups compared to control (WT: 90.7% and KO: 138%) with a 48% increase in the KO group compared to the WT group. P70S6K showed 35% more phosphorylation in the KO group (compared to the WT HFD group) (an arbitrary cut off was set at 30%; Table 3).

Table shows percentage difference in phosphorylation of proteins in both genotypes compared the control group (WT ND, column 3) and then the percentage difference between the two genotypes (column 4).

Phosphorylated P70S6K acts in an inhibitory manner on the insulin receptor adapter protein IRS; PRAS40 phosphorylation leads to dissociation of this inhibitor from mTOR. Because hepatocyte reparative proliferation is decreased in steatosis [54], the increase in ERK phosphorylation is likely a reflection of the development of steatohepatitis in KO, when inflammatory cells invade the tissue (Figure 2a) and suggests increased lipotoxicity in KO. Taken together, these changes are likely to reflect a greater extent of insulin resistance in KO relative to WT and greater abundance of inflammatory cells/hepatomegaly in KO.

### 3.3. C5L2 and CD36 Abundances in Adipose Tissue

Because adipose tissue inflammation precedes hepatic inflammation in mice consuming a high fat diet [55], adipose tissue membrane lysates were analysed for C5L2 and CD36. There was a significant decrease in the expression of C5L2 in the fat membrane of the KO HFD group compared to the WT HFD group (Figure 3a), as expected. No CD36 reactivity was evident in the control group lane or the lanes of the WT HFD group (Figure 3b). 3 of the 4 KO group lanes showed CD36 reactivity.

### 3.4. Circulating C5L2 and CD36

While less CD36 immunoreactivity was present in livers from WT on HFD compared to KO on HFD (Figure 4), detection of C5L2 was not convincing. However, C5L2 reactivity was unexpectedly present in plasma by Western blotting and was also detectable using a commercially available ELISA, though groups were indistinguishable (data not shown).

We next investigated the presence of CD36 in plasma using ELISA. It was found in all groups although expression in the control group was comparatively low. There was a significant increase in CD36 protein in the WT HFD group plasma compared to controls (*p* < 0.0095). The CD36 concentration was higher still in the KO group with a significant difference between the genotypes (*p* < 0.0260) (Figure 5). CD36 in plasma has been described to be associated with microparticles [59]. Therefore, we decided to use flow cytometry of plasma samples to determine whether microparticles [60] might be the origin of C5L2 and CD36 in plasma.

We found that there were more circulating microparticles (MP) in plasma from mice on HFD than from mice on normal diet (Figure 6a). When analysing the MPs further for C5L2 or CD36 positivity, the relative proportions followed the concentrations measured in adipose tissue membrane lysates and plasma, respectively, by ELISA (Figure 3 and Figure 5): There were less C5L2 positive MPs in KO fed a HFD compared to WT, and more MPs stained positive for CD36 in the HFD KO group than the HFD WT group (Figure 6b). Because of the role of small size MPs in promoting cardiovascular disease [61], we measured the percentage of low forward scatter (FSC) type MPs in our pooled samples. HFD KO had about 1.5-fold more low-FSC type MP than HFD WT (Figure 6c).

### 3.5. Characterisation of Obesity Related Renal Pathology

Glomeruli from KO on HFD compared to WT showed greater cellularity and there was more hyaline (proteinaceous) material in tubular epithelium from KO on HFD compared to WT (Figure 7a). Metaplasia of Bowman’s capsule is strain related, but a greater abundance goes along with more resorptive capacity. CD36 IHC gave punctate reactivity in tubules for both, KO and WT (data not shown). Proximal tubular epithelial cells had cytoplasmic vacuoles which appeared round and variably sized (Figure 7b,c) and were more apparent in KO on HFD and may correspond to lipids [62]. Consistent with prior description of obesity-associated glomerulopathy [63,64], we observed fused podocyte foot processes, irregular thickened basal membrane of capillaries and intraglomerular accumulation of lipid droplets in electron micrographs. There was no stark difference in these features between the genotypes on a high fat diet. In the study of the epithelium, however, the presence of mitochondrial autophagosomes was a noticeable feature in KO on HFD, while WT on HFD showed a greater abundance of Type III damaged mitochondria [65] (Figure 7d).

To assign functional importance to the ultrastructural changes observed, void urines were collected and tested for the presence of β2 microglobulin as damage marker of the proximal tubular epithelium [66]. HFD fed KO showed a clear pathological presence of β2 microglobulin (Figure 8). There was progression in the excretion of β2 microglobulin with duration of HFD which was greater for KO (data not shown). Whole kidney lysates were analysed for the abundance of smooth muscle α actin and showed an increase in KO on HFD compared to WT on HFD, indicative of greater mesangial cell proliferation in KO (Figure 8).

Liver, skeletal muscle and adipose tissue are classically regarded as insulin sensitive organs, but insulin also signals in the kidney via PI3K/pAKT pathway [34].

AKT phosphorylation was increased in both HFD groups compared to control. Phosphorylation of all other proteins measured in the array was increased in the WT HFD group compared to control. In the KO group all except 4 of the proteins measured showed a significant increase in phosphorylation compared to control, and one protein (P70S6K) showed a significant decrease compared to control (46.6%). When comparing phosphorylation of proteins between the genotypes in the experimental groups, phosphorylation of most proteins in the WT HFD group was markedly increased compared to the KO HFD group (Table 4).

It is known that insulin signalling via AKT in models of obesity differs qualitatively in liver and kidney [67]. Taking an arbitrary cut off of 35% for changes between the two genotypes on HFD, changes in phosphorylation are consistent with i. greater insulin resistance in KO: relatively decreased ERK, rpS6 and PRAS40 phosphorylations and ii. detrimental proliferative and possibly metabolic response in KO: relatively decreased p27 (cyclin dependent kinase inhibitor protein) and p53 phosphorylations. These changes may predispose the kidney to mesangial proliferation and matrix deposition [68,69,70].

In summary, in the absence of amplification of the alternative pathway of complement, the response to a high fat diet is skewed towards a worse phenotype that affects metabolism and tissue function. Against a genetic background that was not amenable to carry out a glucose tolerance test [71], a change in insulin mediated signalling sensitivity was shown in both liver and kidney by studying AKT dependent phosphorylation. C5L2 and CD36 as fatty acid relevant receptors were shown to be significantly dysregulated in KO fed a high fat diet.

## 4. Discussion

Complement activation and metabolism are areas of overlap. Complement components and pathway activities have been found altered in conditions of metabolic strain. Properdin has been implicated in regulating the metabolic response to a high fat diet. This study shows that consumption of high fat diet for ten weeks led to activation of the alternative pathway in vivo, while the classical pathway was not significantly changed. Our work defines the detrimental role of properdin deficiency in precipitating a prediabetic phenotype.

The worse phenotype of KO was expected because of the relative impairment in ASP production afforded in the absence of properdin. We have previously shown that KO in models of inflammation are impaired in their generation of complement split products [72,73,74,75,76]. Less generation of ASP signifies less stimulation of glucose transport [21] and less inhibition of hormone sensitive lipase [12]. Consistent with these interactions, we found significantly elevated levels of HbA1c as well as worse fatty liver disease, secondary to increased adipose tissue lipolysis in KO on HFD. This is the first study to extend the investigations of complement regulated metabolic response to HFD to include renal pathology. The kidney responds to a high fat diet before the development of type 2 diabetes. Leptin receptor abundance was downregulated in kidney lysates from HFD KO and WT compared to ND (data not shown) and proves that kidneys were responsive to metabolic regulation in our model.

When mice were fed a high fat diet, they showed signs of systemic inflammation (C5a, IL-6, CCL2, IL-1β) [77], consistent with the notion that obesity is a risk factor to develop “a state of chronic low-level inflammation” [78]. Plasma C5a was significantly increased, while adiponectin was concomitantly decreased [77]. This study proposed that complement activation related to the excessive presence of saturated fatty acids in blood [77]. Consumption of a fatty meal and subsequent arrival of chylomicrons, packaged lipoproteins, via lymph (from intestine) and blood to adipose tissue leads to an increased production of C3. In fact, C3 levels associate with body mass indices in obesity [79]. Increased engagement of RAGE (Receptor for advanced glycation end products)/p38MAPK-NF-κB pathway may itself lead to increased production of C3 [80]. However, lipoxidation, the extent to which chronically elevated lipoproteins are oxidatively modified [81], is present and likely confounds identification of the trigger of complement activation [82]. In fact, in obese subjects, levels of C3a desArg/ASP and lipid hydroperoxide were significantly correlated [83].

Adipose tissue, in the state of overfeeding, reverses the flux of fatty acids: While normally triglycerides are synthesised from fatty acids in the presence of insulin, a release of fatty acids is observed in a state of overfeeding, when insulin resistance develops [84]. The extent of the interaction of C3a desArg and C5L2 is likely to co-determine this handling of fatty acids [85]. We have previously observed that properdin deficient mice (mice with abrogated alternative pathway amplification), when fed a Western Diet (45% kcal fat), have decreased C3a desArg levels and were significantly more obese compared to wildtype mice [11]. Cellular accumulation of triglycerides and unesterified fatty acids exert a so-called lipotoxic effect, which contributes to obesity related end organ disease [86]. Increased lipid peroxidation-derived aldehydes and increased oxidative stress in accumulated fat also contribute to the state of so-called metainflammation, which is observed in diet-induced obesity [87,88].

Our study shows that the phenotype of diet-induced obesity and accompanying inflammation, so-called metainflammation, is worse in the absence of properdin.

Circulating fatty acids exert a pro-inflammatory effect, which dampens insulin signalling in liver and adipose tissue via fatty acid induced activation of the inflammasome [89]. HbA1c levels were significantly more elevated in KO on HFD, mirroring the inefficiency of insulin to lower glucose levels. The phosphoarray analyses of livers and kidneys indicated that insulin sensitivity was relatively more compromised in KO on HFD. An increase in CD36 leads to increased uptake of fatty acids; this stimulates cytokine release [90] and in vivo, associates with development of steatosis and systemic insulin resistance [91] as well as inflammation in adipose tissue [92]. Cytokines downregulate C5L2 expression, measured by binding of C3a desArg [93]. C3a desArg/C5L2 interaction decreases lipolysis, but C3a desArg is diminished in KO. In overweight men, lower ASP (C3a desArg) levels associate with the development of metabolic syndrome [94], when fatty acids exert their lipotoxic effect in organs. Release of fatty acids from adipose tissue where it is normally stored in esterified form, is increased during insulin resistance. The increase of CD36 in adipose tissue and liver is likely to be secondary to the developing loss of insulin sensitivity [95,96,97]. Elevated sCD36 levels were shown to be associated with insulin resistance [97]. The metabolic state of increased insulin resistance yields the relative reduction of C5L2 in KO compared to congenic wildtype [98].

It is important to stress that consumption of a high fat diet is relevant for the systemic activation of the alternative pathway of complement. Overall, KO are experiencing greater metainflammation and a worse extent of developing insulin resistance: their reaction to an obesogenic diet is much more profound compared to the congenic wildtype mice in terms of adiponectin levels, hypertriglyceridemia, development of steatosis. If C3a desArg aids in the triglyceride synthesis in adipose tissue, then a relative reduction of this ligand could lead to greater uptake of dietary and released fatty acids in liver. The decline in C5L2 and increase in CD36 in the adipose tissue of KO is mirrored in liver and plasma (microparticles). The renal involvement in response to effects of altered handling of the high fat diet are greater in KO. Accumulation of cytoplasmic lipid droplets has been described in obesity [99] and we have found changes in kidney compatible with this interpretation. The mitochondrial changes we have observed are in line with their pathogenic involvement in obesity-related kidney damage that has previously been described [100].

The work analyses genetically modified mice (which were engineered to be deficient of properdin by site-specific recombination with a disruption construct). Therefore, all change towards diet is secondary to the properdin defect. By reverse conclusion, in congenic wildtype mice (compared to KO mice), intact activity of the alternative pathway of complement contributed significantly to limit diet induced endotoxemia, prediabetic metabolic response and hepatic steatosis.

We propose the following pathomechanism (Figure 9).

A high fat diet changes the microbiome [101] and increases intestinal CD36 abundance [102]. In the small intestines, exposed to high fat diet, the complement transcriptome is increased [103] as epithelial cells react to support the mucosal barrier [104]. In a condition where epithelial integrity is compromised, we found that the absence of properdin led to enhanced bacterial translocation [105]. Therefore, we suggest that altered intestinal permeability in KO (which we did not measure) yields higher levels of endotoxin, fatty acids and lipids (which we measured). Alternative pathway mediated complement activation is reduced, C5L2 is less engaged to allow the appropriate uptake of free fatty acids into adipocytes while CD36 increases in a compensatory manner. A worse metabolic state ensues with the generation of microparticles which mirror the increase in CD36 [60]. The microparticles are pathologically relevant to aggravation of the damage in the liver [32] which converts the increased fatty acids to lipids, suffering lipotoxicity. Diet-induced endotoxemia associates with insulin resistance [106]. How endotoxin signalling in the context of metainflammation is integrated in C5L2 responses [107] however, requires further study. It is likely that the normal, fatty acid induced increase of C5L2 expression is significantly altered at the post-intestinal end organ in the presence of a pro-inflammatory cytokine profile.

## 5. Conclusions

The alternative pathway activation of complement is important in limiting the development of a prediabetic phenotype via the alteration of C5L2 and CD36, while it interfaces with inflammatory and metabolic changes associated with consumption of a high fat diet.

## Figures and Tables

**Figure 1 medicina-56-00484-f001:**
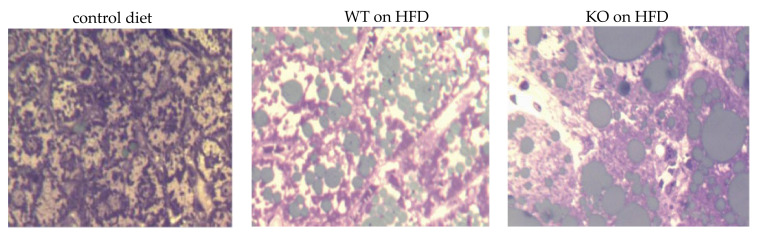
High fat induced steatosis. Semithin sections (400 nm) of epoxy resin embedded mouse livers stained with osmium tetroxide (lipid droplets appear olive green) and counterstained with toluidine blue. ×40 oil.

**Figure 2 medicina-56-00484-f002:**
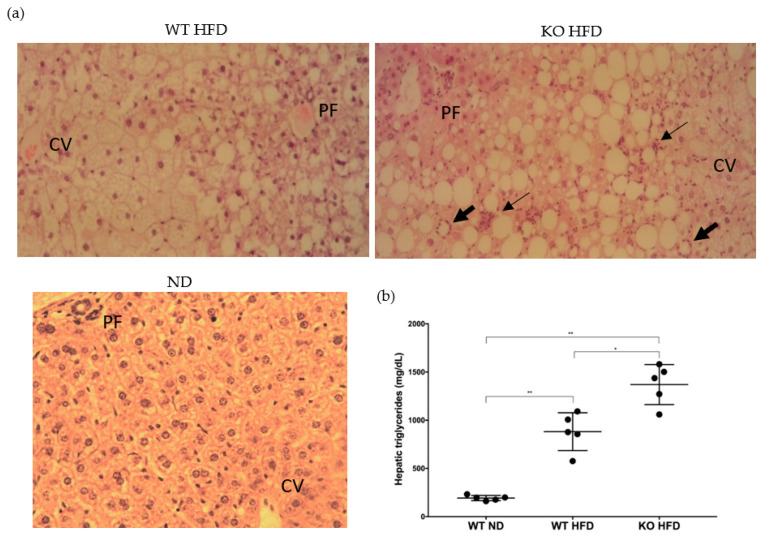
Development of fatty liver disease. Hematoxylin Eosin stained sections of mouse livers showing macrovesicular steatosis and hepatocellular ballooning. Inflammatory cells in groups (thin arrow) and surrounding liver cells (thick arrow). CV, central vein PF, portal field. ×20 magnification (**a**). Triglyceride levels in the livers of C57BL/6J WT and properdin KO mice after consumption of a high-fat diet for 12 weeks and wild type control mice fed on normal chow (*n* = 5 per group). Statistical analysis was performed using the Mann-Whitney U test. Data expressed as mean and standard deviation. * *p* < 0.05; ** *p* < 0.01 (**b**).

**Figure 3 medicina-56-00484-f003:**
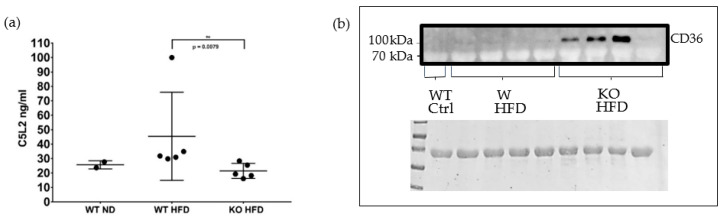
Analyses of adipose tissue membranes for C5L2 and CD36. ELISA for C5L2 in adipose tissue membrane lysates. Fat membrane lysates were prepared from epidydimal fat pads of wild type mice fed on a normal diet (WT ND) and of wild type and properdin knockout mice fed on a high fat diet (WT HFD, KO HFD) for 12 weeks and C5L2 content measured by ELISA. Statistical analysis was performed using the Mann-Whitney U test. Data expressed as mean and standard deviation. ** *p* < 0.01 (**a**). Western blot of adipose tissue membrane lysates for CD36. top panel, Western blot for CD36 in adipose tissue membrane lysates from C57BL/6J wild type (WT) and properdin knockout mice (KO) fed a high-fat diet (HFD) for 12 weeks and a wild type control mouse fed on normal chow. Note, the predicted molecular weight is 53 kDa, however its observed weight (when separated on an SDS PAGE gel) tends to be around 78–94 kDa [56,57] due to the post translational addition of glycans which are necessary for its translocation to the plasma membrane [58]. Bottom panel, Ponceau stain of the membrane immediately following transfer (**b**).

**Figure 4 medicina-56-00484-f004:**
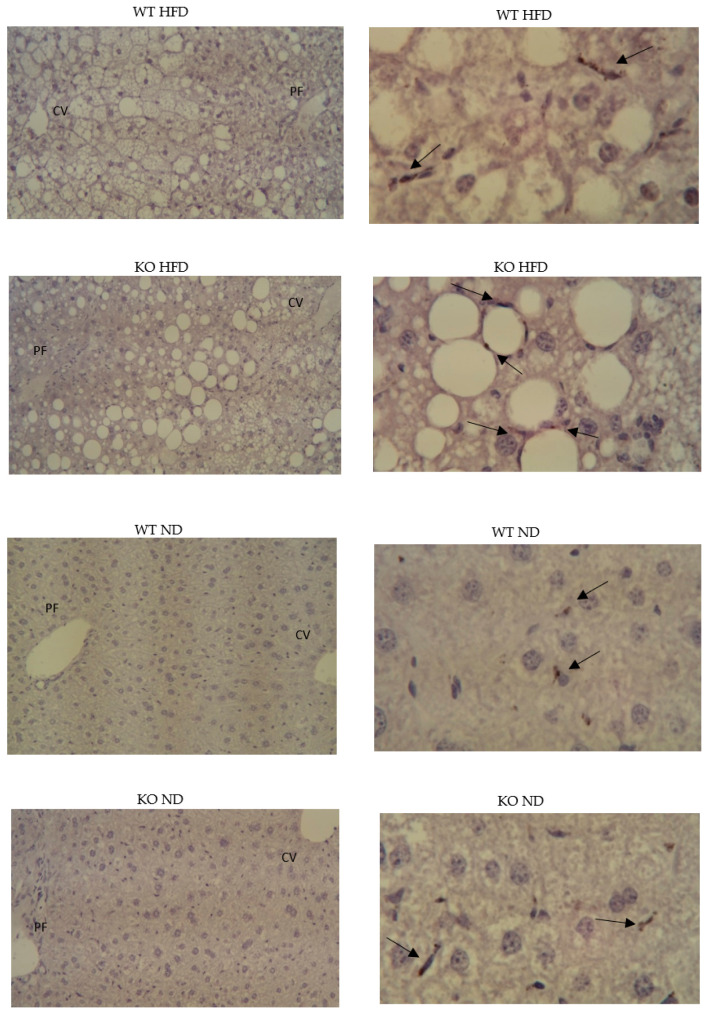
Hepatic presence of CD36. left panels ×10 (negative controls); right panels ×40 oil, CD36 immunoreactivity. Paraffin embedded, hematoxylin stained.

**Figure 5 medicina-56-00484-f005:**
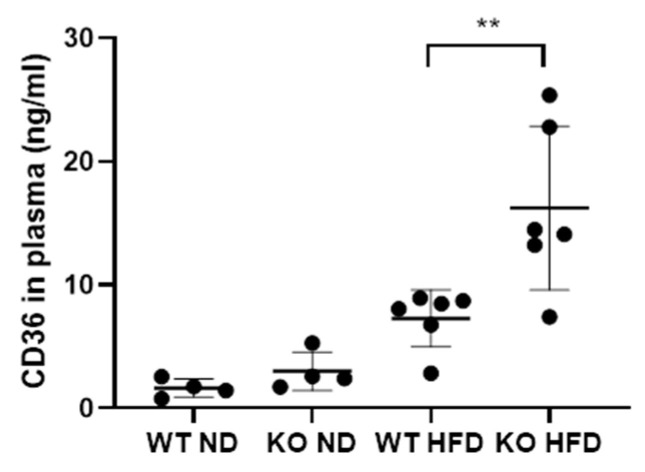
Concentration of CD36 in the plasma of wild type mice fed on a normal diet (WT ND) properdin knockout mice fed on a normal diet (KO ND) wild type mice fed on a high-fat diet (WT HFD) and properdin knockout mice fed on a high fat diet (KO HFD) for 12 weeks. Statistical analysis was performed using one-way ANOVA with Tukey’s multiple comparison test. Data expressed as mean and standard deviation. ** *p* < 0.01. *n* = 4 per group for control mice and 6 per group for high-fat diet fed mice.

**Figure 6 medicina-56-00484-f006:**
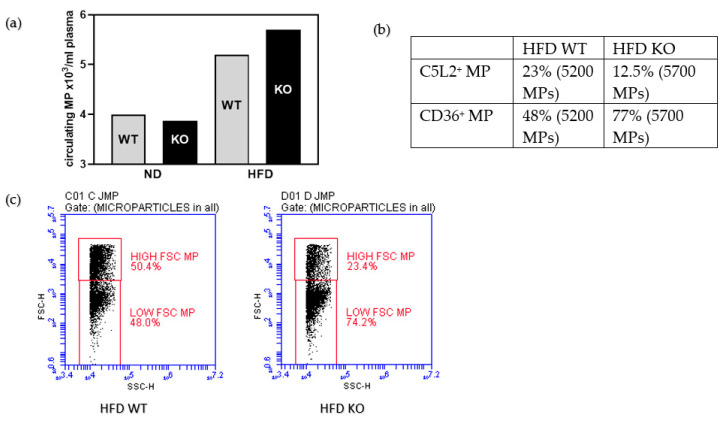
In a pool of plasma from five mice each genotype enumeration of all microparticles, MP (**a**), proportion of C5L2 or CD36 reactivities of all microparticles measured (**b**), percentage of low FSC small microparticles (**c**).

**Figure 7 medicina-56-00484-f007:**
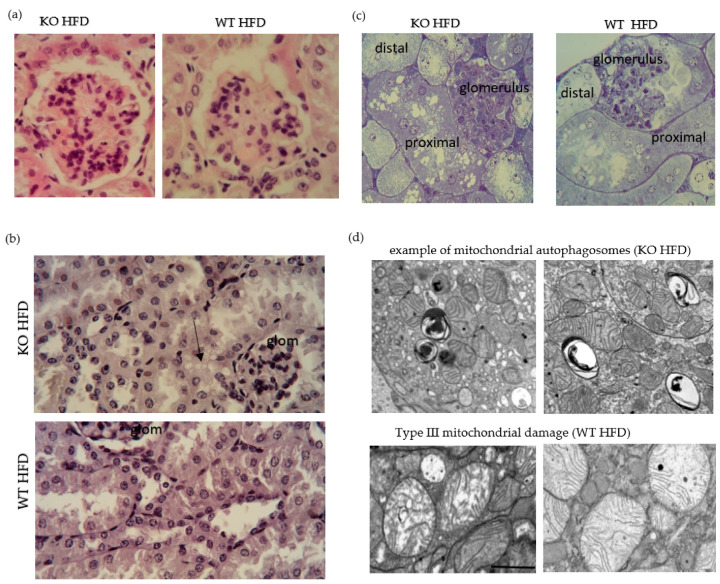
High fat diet induced renal histopathology. Representative image of glomeruli from HFD fed WT and KO; hematoxylin and eosin stain ×40 oil. Focal segmental proliferation in KO. Note greater pink stain in epithelial stain in KO. Epithelisation of Bowman’s capsule is described in mouse (**a**). Bismarck Brown stained. Note the presence of vacuoles in epithelium adjacent to the glomerulus (arrow) (**b**). Epoxy resin embedded, semithin sections, toluidine blue stained. Note greater abundance of vacuoles in epithelium of proximal tubule (**c**). Electron micrographs showing typical features observed in KO and WT on HFD. scale bar is one micron (**d**).

**Figure 8 medicina-56-00484-f008:**
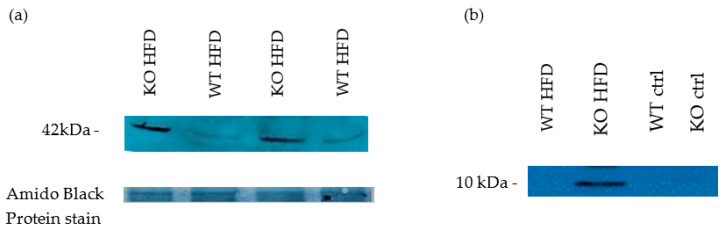
Western blot for Markers of kidney impairment. 8 weeks HFD. Smooth muscle alpha actin (profibrotic marker protein- mesangial cell proliferation marker). Kidney lysates (NP40) (**a**) Void urine samples, probed for β2 microglobulin as proximal tubular epithelium-damage marker due to presumed increased resorption load 8 weeks HFD (**b**).

**Figure 9 medicina-56-00484-f009:**
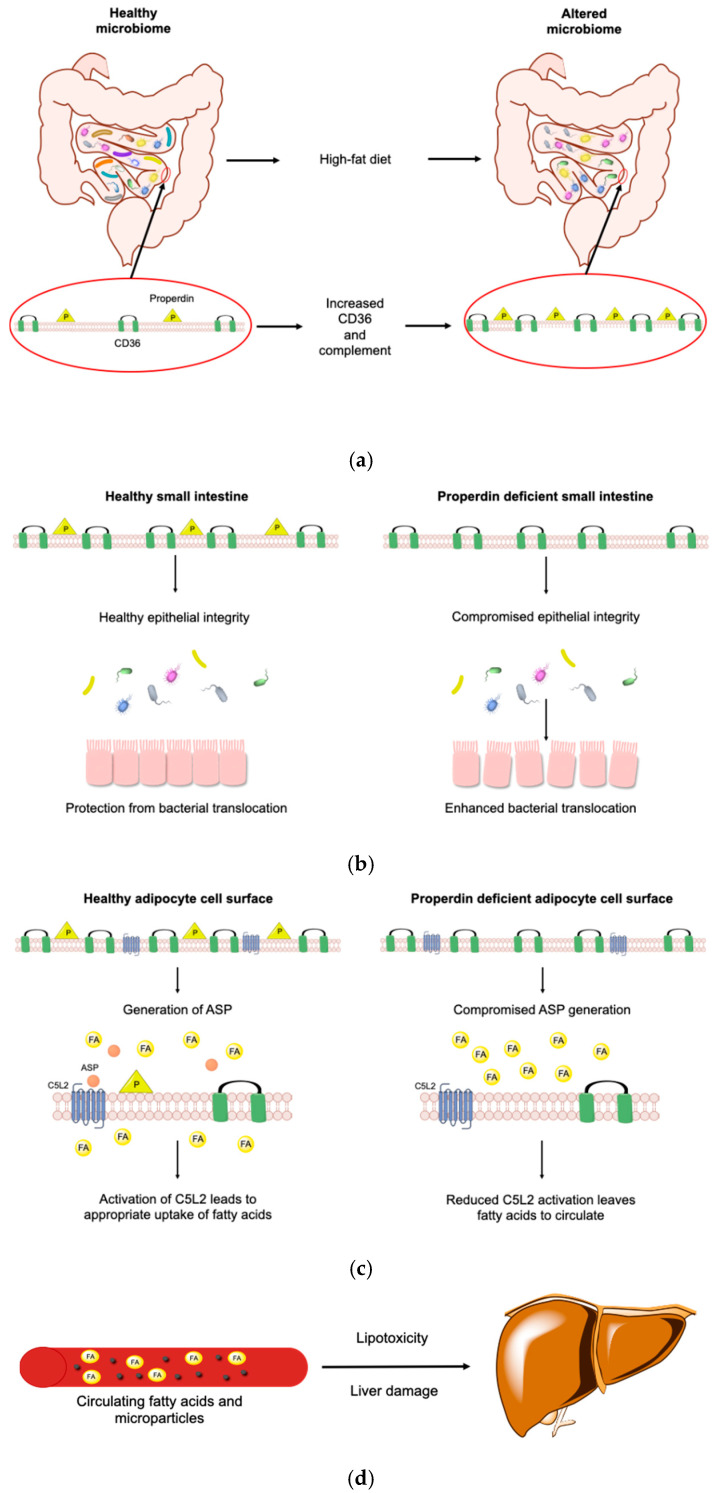
Pathomechanism of the properdin modulated obese phenotype. A high fat diet changes the microbiome and increases intestinal CD36 abundance (**a**). High-fat diet increases the complement transcriptome in the small intestine (left). Absence of properdin leads to enhanced bacterial translocation (right) and compromised epithelial integrity (**b**). During exposure to a high fat diet (right and left) alternative pathway mediated complement activation is reduced (right) so C5L2 is less engaged to allow the appropriate uptake of free fatty acids into adipocytes while CD36 increases in a compensatory manner (**c**). Increased circulating fatty acids are taken up by the liver and a worse metabolic state ensues with the generation of microparticles resulting in damage to the liver which is also converting the increased fatty acids to lipids, suffering lipotoxicity (**d**). (Illustration made with ChemDraw Professional).

**Table 1 medicina-56-00484-t001:** Biochemical and inflammatory markers in properdin-deficient (KO) and wild type (WT) mice fed a high-fat diet (HFD).

Parameter (Expressed as Means ± SD)	HFD WT (*n* = 3)	HFD KO (*n* = 3)	Unpaired *t*-Test
HbA1c * (fmol/L)	42.95 ± 1.04	76.06 ± 4.58	*p* < 0.0005
Adiponectin (ng/mL)	1.90 ± 0.14	1.53 ± 0.1	*p* < 0.02
Triglycerides (mg/dL)	23.11 ± 1.70	41.55 ± 5.93	*p* < 0.01
NEFA (mmol/L)	0.37 ± 0.1	0.59 ± 0.02	*p* < 0.005
Endotoxin (IU/mL)	4.53 ± 0.78	13.92 ± 4.12	*p* < 0.02
Malondialdehyde (nmol/mL)	0.27 ± 0.01	0.32 ± 0.02	*p* < 0.01
IL-6 (ng/mL)	8.13 ± 1.71	14.67 ± 1.33	*p* < 0.01
C5a (ng/mL)	321.5 ± 10.61	212.6 ± 12.16	*p* < 0.02
ALT (IU/L)	65.8 ± 9.45	109.1 ± 12.88	*p* < 0.01

* HbA1c, glycosylation end product that mirrors past glucose control over 40 days.

**Table 2 medicina-56-00484-t002:** Relative complement activity levels in pooled serum from experimental mice (*n* = 3 each), for alternative and classical pathways of activation.

Genotype, Diet	Pathway Tested	Activity Relative to NMS *
WT ND	Alternative pathway	89%
KO ND	Alternative pathway	6%
WT HFD	Alternative pathway	27%
KO HFD	Alternative pathway	n.d.
WT ND	Classical pathway	80%
KO ND	Classical pathway	40%
WT HFD	Classical pathway	72%
KO HFD	Classical pathway	47%

n.d., not done. Heat lability and EDTA controls as well as background absorbance were carried out in parallel. * NMS, commercial normal mouse serum.

**Table 3 medicina-56-00484-t003:** Phosphorylation of proteins of the AKT pathway, downstream of the insulin receptor in liver.

Phosphorylated Protein	Genotype	% Difference from Control	% Difference between Genotypes
ERK 1/2	WT KO	+176.3	62
		+238	
PRAS40	WT KO	+90.7	48
		+138.8	
PTEN	WT KO	+69.6	11
		+80.2	
GSK3α	WT KO	+98.4	4
		+94.3	
GSK3β	WT KO	+101.0	12
		+79.9	
Raf-1	WT KO	+77.2	12
		+89.4	
mTOR	WT KO	+70.6	21
		+58.4	
AKT	WT KO	+162.5	91
		+71.4	
RPS6	WT KO	+65.9	13
		+78.7	
P27	WT KO	+38.0	20
		+57.9	
RSK1	WT KO	+43.7	10
		+53.5	
RSK2	WT KO	+64.5	10
		+74.1	
AMPK	WT KO	+57.0	11
		+45.9	
P53	WT KO	+68.3	23
		+91.0	
BAD	WT KO	+40.5	27
		+67.9	
P70S6K	WT KO	+138.5	35
		+173.5	
PDK1	WT KO	+50.4	0
		+50.8	

**Table 4 medicina-56-00484-t004:** Phosphorylation of proteins of the AKT pathway, downstream of the insulin receptor in kidney.

Phosphorylated Protein	Genotype	% Difference from Control	% Difference between Genotypes
ERK 1/2	WT KO	+62.0	48
		+13.9	
PRAS40	WT KO	+112.9	76
		+37.2	
PTEN	WT KO	+57.7	22
		+36.1	
GSK3α	WT KO	+72.2	23
		+49.2	
GSK3β	WT KO	+76.0	25
		+51.0	
Raf-1	WT KO	+59.7	33
		+26.5	
mTOR	WT KO	+70.6	30
		+40.3	
AKT	WT KO	+57.3	8
		+64.8	
RPS6	WT KO	+55.5	41
		+14.6	
P27	WT KO	+43.0	59
		−16.4	
RSK1	WT KO	+41.7	34
		+7.7	
RSK2	WT KO	+25.3	26
		−0.3	
AMPK	WT KO	+44.9	13
		+58.0	
P53	WT KO	+36.6	39
		−2.4	
BAD	WT KO	+47.8	7
		+41.2	
P70S6K	WT KO	+14.2	32
		−46.6	
PDK1	WT KO	+35.9	16
		+20.0	

Table shows the percentage difference in phosphorylation of proteins in both genotypes compared the control group (wild type normal diet-fed animals, column 3) and then the percentage difference between the two genotypes (column 4).

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
