# Peer review of "Complement Properdin Regulates the Metabolo-Inflammatory Response to a High Fat Diet"

_medicina, 2020, doi:10.3390/medicina56090484_

Round 1

Reviewer 1 Report

Dear authors,

greatings for your work! I found the article very interesting and innovative. In my literature review i didn't find articles about properdin involvment in metabolic activity.

Just same minor advice:

  • i suggest to improve the text readability, in particular regarding the introduction.
  • did you study on your murine model changes in epicardial or perirenal fat? these kind of adipose tissue have a well known role in cardiovascular risk factor and pro-inflammatory action (Dozio E, Ruscica M, Vianello E, et al. PCSK9 Expression in Epicardial Adipose Tissue: Molecular Association with Local Tissue Inflammation. Mediators Inflamm. 2020;2020:1348913.)
  • typical features of obesity related nephropaty (glomerulomegalia, focal or diffuse sclerosis etc.) seems to be not present in your models. What about that? (Xu T, Sheng Z, Yao L. Obesity-related glomerulopathy: pathogenesis, pathologic, clinical characteristics and treatment. Front Med. 2017;11(3):340-348.)

Best regards!

Author Response

Thank you for your review and comments. 

We have reworked the introduction and discussion to make the text more readable. 

Our model did not lead to a significant accumulation of epicardial fat, but there were certainly large deposits in the perirenal area. We agree, we would have liked to have analysed perirenal fat, had more funding been available, especially as a role for renal lipotoxicity and in sustaining detrimental inflammation has recently been reviewed (Huang et al. Novel insight into perirenal adipose tissue: A neglected adipose depot linking cardiovascular and chronic kidney disease. World J Diabetes. 2020. doi:10.4239/wjd.v11.i4.115).  

Variables in the expression of obesity related glomerulopathy are the species that are studied, duration of metabolic dysregulation, the aggressiveness of the diet, the strains of experimental mice. These influence the humane endpoints of the published studies and limit comparability between these works as well as the scope to translate to human disease. 

Reviewer 2 Report

Complement properdin regulates the metabolo3 inflammatory response to a high fat diet Rόisín C. Thomas1*, Ramiar Kheder1**, Hasanain Alaridhee1 , Naomi Martin1,2 , Cordula M. Stover 4 1 

P2

31 The C5L2 (C5aR2 or GP77) receptor is a seven transmembrane receptor that is expressed in a

32 diverse array of cell types and tissues including immune cells, adipose tissue and the liver [10]. It is

33 similar in structure to the C5a complement receptor (C5aR) but unlike the C5aR it does not couple to

C5aR should be C5aR1

34 (and therefore does not signal via) Gα-proteins. It was initially thought to function only as a decoy

35 receptor, due to its ability to bind C5a and C5a DesArg (a metabolite of C5a) and subsequently

36 undergo internalisation, leading to degradation of those ligands [14].

I don’t think metabolite is the best term to describe C5aDesArg. “cleavage product of C5a” perhaps?

P5

21 Equal amounts of tissue were placed into ice cold 1% NP40 lysis buffer

Insert (v/v)

35 skimmed milk and 0.1% (v/v) Tween 20 before incubation overnight at 4°C in PBS/T-5% milk

Insert (w/v)

38 PBS/T-5% milk

Insert (w/v)

P6

22 Gaussian distribution of measurements was assumed.

Why was this “assumed” and not tested?

38 Table 1

Parameter (expressed as means +/- SD) HFD WT (n=3) HFD KO (n=3)     Unpaired t-test

Endotoxin (IU/ml)                                            4.53+/-0.78     13.92+/-4.12 p<0.02

Interesting…so is the gut epithelium more leaky in the properdin KO mice on a hfd? And is this accounting for much of the systemic inflammatory effects? What are the endotoxin levels like in KO mice on normal chow vs wt on normal chow? Was this done?

P7

The normal activity in

21 serum from WT mice was significantly decreased when serum from mice fed a high fat diet was

22 tested. This is consistent with increased activation in vivo, leading to consumption of the complement

23 components necessary to yield deposited C9 (in the form of C5b-poly C9) on the plate.

WT HFD Alternative pathway 27%

Interesting! Large consumption of alternative pathway in mice on HFD.

Fat collecting in liver not adipose tissue….. due to complement dysregulation.

P9

Table 3, do these arrays allow generation of Statistical tests?

P10

Fig 3

I am worried about the outlier in the WT HFD group but by doing a crude calculation reading numbers off graph this looks not to be a problem (ie Mann Whitney is showing significant difference)

P11

Fig 4 arrows would help to point out specific staining.

Fig 5. Is the increase in CD36 indicative of trying to compensate for los of “normal” lipid clearance pathways through C3aDesArg> C5L2?

Fig 7 would benefit from some e.m. taken showing tissues from animals on a ND

P14

Table 4

Do these arrays allow generation of Statistical tests?

Discussion

Good discussion covering most of my questions. It is clear now from a number of studies (both human and animal) that complement plays an important role in daily lipid clearance and metabolism and that activation is a normal, common, post prandial event.

In the apoE model we have seen that increased alternative pathway activation (in absence of CD55 [DAF]) lowers triglyceride levels, increases amount of adipose tissue and decreases atherosclerosis. (This is not the case in the LDLr/DAF where perhaps the absence of the receptor nullifies any positive impact of increased complement activation). However, one might hypothesise that over a long period (years in humans) of increased fat intake such protective mechanisms could be overcome leading to the the chronic problems associated with obesity, including diabetes and heart disease.

Author Response

We thank you for your detailed comments and address the points of critique as follows.

C5aR has been changed to C5aR1 in accordance with the literature.  C5adesarg has been identified as hydrolytic product (because of carboxypeptidase activity). 

v/v and w/v have been inserted where they were missing. 

Gaussian distribution was assumed: For the different group sizes tested there was no reason why individual measurements among the two genotypes would not distribute according to Gauss.  

Our discussion elaborates the conclusion that KO on HFD present with a leakier gut epithelium. Endotoxin levels were determined in mice on normal chow: these were less than 2 IU/ml, and genotypes were indistinguishable.  

Tables 3 and 4: The array data were expressed as described in M&M; we worked with pools of experimental tissue, normalised intensities to normal chow fed controls and then used an algorithm as per manual that reveals the fold-change/percentage increase or decrease in the diet groups. We were stringent in our interpretation by working with relatively high arbitrary cut offs as indicated. 

Figure 3: Without outlier the p-value is 0.0159 and with outlier is 0.0079 so yes, it is still significant.

Figure 4: arrows have been added as requested, to identify exemplary CD36 reactivity.

Figure 7: The electron micrographs were scanned for published criteria of mitochondrial damage to provide further ultrastructural support for the staged detrimental renal involvement in KO over WT on HFD. This is why normal mice were not included in this analysis. 

It is difficult to compare different disease models solely from the point of view of the role complement plays; ApoE deficient models show poor lipoprotein clearance so the mechanism studied differs from a high fat diet overload model. We have reviewed  the role of complement and studied the role of properdin in atherosclerosis previously, and would like to refer to our papers: 

Francescut et al. The role of complement in the development and manifestation of murine atherogenic inflammation: novel avenues. J Innate Immun. 2012;4(3):260-272. doi:10.1159/000332435

Steiner et al. Protective role for properdin in progression of experimental murine atherosclerosis. PLoS One. 2014;9(3):e92404. doi:10.1371/journal.pone.0092404